# Curvature MPNNs : Improving Message Passing with Local Structural Properties

## Abstract

Graph neural networks follow an iterative scheme of updating node representations based on the aggregation from nearby nodes known as the message passing paradigm. Although they are widely used, it has been established that they suffer from a problem of over-squashing that limits their efficiency. Recently, it has been shown that the bottleneck phenomenon comes from certain areas of the graphs, which can be identified by a measure of edge curvature. In this paper, we propose a framework appropriate for any MPNN architecture that distributes information based on the curvature of the graph's edges. Experiments conducted on different datasets show that our method mitigates over-squashing and outperforms existing graph rewiring methods in several node classification datasets.

## 1 Introduction

Graph representation learning is a rapidly expanding research field that focuses on the development of versatile methods for effectively learning representations from graph-structured data (Goller & Kuchler, 1996) (Gori et al., 2005) (Scarselli et al., 2008) (Bruna et al., 2014). The majority of Graph neural networks *GNNs* are based on the message passing paradigm (Gilmer et al., 2017), in which the information is propagated by the iterative exchange of information (messages) between neighboring nodes to update their representations. This process is typically performed over multiple iterations and/or layers. The message passing paradigm is effective in capturing the relational information and structural patterns within graph-structured data. It enables GNNs to learn expressive representations that are sensitive to the connectivity and interactions among nodes in a graph. GNNs have been successful in various domains, including chemistry, information retrieval, social network analysis, and knowledge graphs, due to the wide variety of features that a graph can model (Wu et al., 2020). These architectures have produced very interesting results when it comes to solving tasks at the node, graph, or edge level (Xiao et al., 2022) (Errica et al., 2020) (Zhang & Chen, 2018).

Despite their widespread use, it has been shown that GNNs can face a variety of issues under certain conditions, specifically in heterophilic environments (Zhu et al., 2020) (Platonov et al., 2023), when the neighboring nodes tend to have different labels. Other works have highlighted that GNNs suffer from a limited ability to model long-range interactions (Alon & Yahav, 2021). Popular GNN architectures such as Graph Convolutional Networks (GCN) (Kipf & Welling, 2017) and Graph Attention Networks (GAT) (Veličković et al., 2018) can only share information between nodes at a distance that depends on the number of layers in the architecture: for a node $i$ to be able to exchange information with a node $j \in \mathcal{N}_k(i)$, we need to stack at least $k$ layers. Therefore, a naive approach to address this issue consists of increasing the number of layers.

However, this process leads to two well-known problems for GNN. First, the phenomenon of over-smoothing which arrives when the message passing is carried out in an excessive way. In this case, all the features of the nodes are going to be similar wich leads to a deterioration in results Oono & Suzuki (2020) Cai & Wang (2020). Second, as the number of layers in a GNN grows, information from exponentially growing receptive fields must be propagated concurrently at each message-passing step, leading to a bottleneck that causes over-squashing (Alon & Yahav, 2021). In this case, spreading information locally is not enough. To overcome this problem, GNNs must

be able to incorporate additional global graph features in the process of learning representations. Another popular approach is to rewire the input graph to improve the connectivity and avoid over-squashing. Recently, it has been shown that the local structural properties of a graph, like edge curvature, play an essential component in the spread of knowledge about the graph (Topping et al., 2022).

**Main Contributions**   This paper presents a new framework for any MPNN architecture to prevent over-squashing. The main contributions are:

- We present a new measure of homophily based on edge curvature that allows us to better model the community behavior of a neighborhood.

- Motivated by this metric we propose a new MPNN model (curvature-constrained message passing) that leverages the curvature of the edges to guide learning by dissociating edges with positive and negative curvature. We propose different variants of this model, each one based on a different way of propagating the information: only on edges with negative curvature, positive curvature, or a combination of both. We also propose two- or one-hop propagation strategies that are bound to the curvature.

- We empirically demonstrate a performance gain on heterophilic datasets and we show that using a curvature message passing attenuates over-squashing.

## 2   RELATED WORK

### 2.1   MESSAGE PASSING NEURAL NETWORKS

The success of deep learning in the Euclidean domain prompted a great interest to generalize neural networks to non-Euclidean domains e.g. graphs. Let $G = (V, E)$ be a simple, undirected, and connected graph with node features $h_i \in R^d, i \in V$. $\mathcal{N}(i)$ represents the set of neighbors of node $i$.

The main objective of the message passing approach is to iteratively find an effective node embedding that captures context and neighborhood information (Gilmer et al., 2017). The message passing technique consists of two phases which iteratively apply the AGGREGATE and UPDATE function to compute embeddings $h_i^\ell$ at the layer $\ell$ based on message $m_i^{(\ell)}$ containing information on neighbors :

$$
\begin{aligned}
m_i^{(\ell)} &= \text{AGGREGATE}^{(\ell)} \left( h_i^{(\ell-1)}, \left\{ h_j^{(\ell-1)} \; j \in \mathcal{N}(i) \right\} \right), \\
h_i^\ell &= \text{UPDATE}^{(\ell)} \left( h_i^{(\ell-1)}, m_i^{(\ell)} \right)
\end{aligned}
\tag{1}
$$

For GCN (Kipf & Welling, 2017) $m_i^\ell = \sum_{j \in \mathcal{N}(i)} \frac{\mathbf{h}_j^\ell}{\sqrt{|\mathcal{N}(i)||\mathcal{N}(j)|}}$ while for GAT (Veličković et al., 2018) $m_i^\ell = \sum_{j \in \mathcal{N}(i)} a_{ij}^\ell \, \mathbf{h}_j$ .

with

$$
a_{ij}^{(\ell)} = \frac{\exp \left( \text{LeakyReLU}(z^{(\ell)} \cdot [h_i^{(\ell-1)} \mathbf{W}^{(\ell)} || h_j^{(\ell-1)} \mathbf{W}^{(\ell)}]) \right)}{\sum_{j \in \mathcal{N}_i} \exp \left( \text{LeakyReLU}(z^{(\ell)} \cdot [h_i^{(\ell-1)} \mathbf{W}^{(\ell)} || h_j^{(\ell-1)} \mathbf{W}^{(\ell)}]) \right)}
$$

where $||$ stands for concatenation. This score is parameterised by $z^{(l)}$ and $\mathbf{W}^{(l)}$, respectively a weight vector and a linear transformation.

As classical MPNNs only send messages along the edges of the graph, this will prove particularly interesting when adjacent nodes in the graph share the same label (homophilic case). On the other hand, working in a heterophilic environment with classical MPNNs can leads to low performance (Zheng et al., 2022). Indeed one of the main drawbacks of classical MPNNs is to rely only on one-hop message propagation. Additional layers must be stacked to capture non-local interactions. However, this leads to over-squashing discussed in the section 2.3.

## 2.2 GRAPH CURVATURE

As for a manifold, the notion of curvature is a good way to classify the local behavior of a graph. Discrete graph curvature describes how the neighbors of two nodes are structurally connected. Forman (2003) and Ollivier (2007) were the first to propose a measure of discrete graph curvature. Numerous studies have demonstrated the usefulness of edge curvature for various graph tasks. For instance (Jost & Liu, 2014) (Ni et al., 2019) (Sia et al., 2019) use Ollivier curvature for community detection.

Ye et al. (2019) have defined Curvature Graph Neural architecture which calculates an attention mechanism based on Ollivier curvature. They demonstrate the benefits of such an architecture for the task of node classification. More recently, Jost & Liu (2014) and Topping et al. (2022) proposed extensions to Forman's curvature to improve its expressiveness. Topping et al. (2022) demonstrate the correlation between edge curvature and over-squashing phenomenon. In this paper, we focus on Ollivier curvature (Ollivier, 2007) and on Augmented Forman Curvature as detailed in Section 3.1.

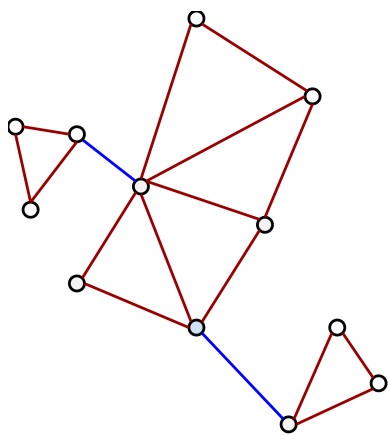

Figure 1: In red, edges with positive curvature,connect nodes in the same community, and in blue, edges with negative curvature connect nodes in different communities.

## 2.3 OVER-SQUASHING

Long-range tasks need the propagation of information across several levels. The node representations are aggregated with others at each stage before being passed on to the next node. Because the size of the node feature vectors remains constant, they rapidly exhaust their representational capacity in order to retain all of the previously integrated information. When an exponentially expanding quantity of information is squashed into a fixed-size vector, over-squashing happens (Alon & Yahav, 2021).

To quantify this phenomenon, some approaches exploit the spectral gap (Banerjee et al., 2022) (Karhadkar et al., 2023), which is closely linked to the Cheeger constant (Chung & Graham, 1997).

$$Ch(G) = \min_{1 \le |S| \le \frac{|V|}{2}} \frac{|\partial S|}{|S|}, \tag{2}$$

With $S \subset V$ and where $\partial S = \{(i,j) : i \in \mathcal{S}, j \in \overline{\mathcal{S}}, (i,j) \in E\}$

If Cheeger's constant is small, there is a bottleneck structure in the sense that there are two large groups of vertices with few edges connecting them. Cheeger's constant is large if a feasible vertex split into two subsets has "many" edges between these two subsets.

Calculating the precise value of $Ch(G)$ is too costly. The discrete Cheeger inequality Alon & Milman (1984) Cheeger (1970) shows the link between the spectral gap and the Cheeger constant. We denote $A$ the adjacency matrix and $D$ the diagonal degree matrix and let $L = I - D^{-1/2}AD^{-1/2}$ be the normalized Laplacian of $G$. The spectral gap of $G$ is the difference between the first two eigenvalues $\lambda_2$ - $\lambda_1$ of L with $\lambda_1 = 0$.

$$\frac{\lambda_2}{2} \le Ch(G) \le \sqrt{2\lambda_2} \tag{3}$$

To mitigate the over-squashing phenomenon different works have proposed various methods to improve the local connectivity of the graph.

**Rewiring methods**   Most methods address over-squashing by rewiring the input graph i.e. modifying the original adjacency matrix such that it has fewer structural bottlenecks. Alon & Yahav (2021) were the first to highlight the problem of GNN over-squashing. They propose to modify the GNN's last layer in order to connect all of the nodes. Topping et al. (2022) shows that the highly negatively curved edges are characteristic of the bottleneck phenomenon and therefore disrupt message passing. They propose a stochastic discrete Ricci Flow (SDRF) rewiring approach, which tries to raise the balanced Forman curvature of negatively curved edges by adding and removing edges. Karhadkar et al. (2023) propose an algorithm (FOSR) for adding edges at each step to maximize the spectral gap. Because calculating the spectral gap for each edge addition is costly, FOSR employs a first-order spectral gap approximation based on matrix perturbation theory.

Without the direct objective of reducing the phenomenon of over-squashing, other methods such as Klicpera et al. (2019) modify the adjacency matrix to improve the connectivity of the graph (DIGL). This method adds edges based on the PageRank algorithm, followed by sparsification. As PageRank works using random walks, DIGL tends to improve the connectivity among nodes in the intra-community parts of the graph.

**Master node**   Another way to reduce over-squashing consists of the introduction of a sort of "global context" by introducing a master node. This node is connected to all other nodes in the graph (Battaglia et al., 2018) (Gilmer et al., 2017). Since the hop distance between all other nodes is at a maximum of two, the reduction of over-squashing is assured (except for the master node). However, in large graphs, incorporating information over a very large neighborhood leads to poor quality of the master node embedding.

**Expander Graphs**   Deac et al. (2022) adopt a strategy based on expander graphs, adding to the GNN a layer based on a Cayley graph of the same size as the original input graph. These graphs have some desirable properties, such as being sparse and having a low diameter. The smaller diameter means that any two nodes in the graph can be reached in a reduced number of hops, which removes bottlenecks.

## 3   CURVATURE MESSAGE PASSING

### 3.1   OLLIVIER CURVATURE

We propose to use two notions of curvature on edges for information diffusion : Ollivier curvature and the Augmented Forman Curvature (Samal et al., 2018). We recall the definition of the Ollivier curvature. Let's define a probability distribution $\mu_i$ over the nodes of the graph in such a way we apply to each node $i$ a lazy random walk probability measure $\alpha$ :

$$\mu_i : j \mapsto \begin{cases} \alpha & if \quad j = i \\ (1-\alpha)/d_i & if \quad j \in \mathcal{N}(i) \\ 0 & \text{otherwise} \end{cases}, \tag{4}$$

Following previous work (Ni et al., 2015) (Ni et al., 2018) we choose $\alpha = 0.5$. We then consider the Wasserstein distance of order 1, $W_1(i,j)$, corresponding to the optimal transport of probability masses from $i$ neighbors to $j$ neighbors.

$$W_1(\mu_i, \mu_j) = \inf_{\alpha \in \Pi(\mu_i, \mu_j)} \sum_{i,j \in V} dist(i,j) M(i,j) \tag{5}$$

where $\Pi(\mu_1, \mu_2)$ denotes the set of probability measures with marginals $\mu_i$ and $\mu_j$. where $M(i,j)$ is the amount of mass moved from i to j along the shortest path of i and j. Finally, the Ollivier curvature $c_{ij}$ of an edge $e_{ij}$ can be defined as :

$$c_{ij} = 1 - \frac{W_1(\mu_i, \mu_j)}{dist(i,j)}, \tag{6}$$

where $dist(i,j)$ is the shortest path beetween node $i$ and node $j$.

## 3.2 Augmented Forman Curvature

The curvature measure proposed by Samal et al. (2018) proposes to extend Forman's curvature taking into account the triangles in the graph to make it more expressive

For an undirected graph

$$c_{ij} = 4 - D_{ii} - D_{jj} + 3m \tag{7}$$

where m is the number of triangles contained in $e_i j$.

## 3.3 Curvature-Constrained Homophily

The homophily of a graph has a determining role in the efficiency of architectures on a node classification task. Many homophily measures exist in literature (Pei et al., 2020) (Zhu et al., 2020) (Lim et al., 2021) (Platonov et al., 2022); the most commonly used are node homophily Pei et al. (2020) which computes the average of the proportion of neighbors that have the same class $y$ for each node and edge homophily $\beta$ Zhu et al. (2020) which corresponds to the fraction of edges that connect nodes of the same class:

$$\beta = \frac{|\{(i,j) : (i,j) \in E \wedge y_i = y_j\}|}{|E|}$$

The main limitation of this measure is that it doesn't fully capture the local structural characteristics of the graphs. Therefore, we propose a new measure of homophily that takes into account the curvature of edges such that:

$$\beta^+ = \frac{|\{(i,j) : (i,j) \in E^+ \wedge y_i = y_j\}|}{|E^+|}$$

Where $E^+$ is the set of edges $(i,j)$ such that $c_{ij} \geq \epsilon$. $\beta^-$ is conversely defined using $E^-$, the set of edges $(i,j)$ such that $c_{ij} < \epsilon$. A high value for the positive curvature homophily means that the fraction of edges that connect nodes within the same community tend to have the same label.

The values of $\beta^+$ and $\beta^-$ obtained for the datasets are shown in Table 6 (see Table 2 for the details of the datasets). We consider the Ollivier-curvature-constrained homophily, so we fix $\epsilon = 0$, for one-hop and two-hop neighborhoods. We also provide the max homophilic gain relative to the initial metrics (Zhu et al., 2020). Note that Augmented Forman-curvature-constrained homophily is provided at appendix A.2.

|  | Dataset | $\beta$ | $\beta^+$ | $\beta^-$ | 2-hop$\beta^+$ | 2-hop$\beta^-$ | Max Homophilic Gain - |
|---|---|---|---|---|---|---|---|
| Heterophilic | Squirrel | 0.23 | 0.28 | 0.29 | 0.25 | 0.29 | 25% |
|  | Chameleon | 0.26 | 0.29 | 0.28 | 0.32 | 0.23 | 24% |
|  | Texas | 0.31 | 0.44 | 0.43 | 0.59 | 0.47 | 92% |
|  | Wisconsin | 0.36 | 0.44 | 0.50 | 0.49 | 0.38 | 37% |
|  | Cornell | 0.34 | 0.48 | 0.46 | 0.37 | 0.40 | 40% |
|  | R-empire/ | 0.29 | 0.40 | 0.48 | 0.05 | 0.07 | 65% |
|  | Actor | 0.32 | 0.73 | 0.32 | 0.32 | 0.21 | 131% |
| Homophilic | Cora | 0.84 | 0.95 | 0.83 | 0.91 | 0.74 | 11% |
|  | Citeseer | 0,81 | 0.86 | 0.84 | 0.80 | 0.75 | 6% |
|  | Photo | 0.84 | 0.94 | 0.73 | 0.94 | 0.54 | 12% |
|  | Computers | 0.78 | 0.93 | 0.70 | 0.94 | 0.59 | 20% |

Table 1: Comparison of edge homophily measures. The last column reports the max gain in homophily obtained by using the curvature-constrained edge homophily as opposed to edge homophily.

For homophilic datasets, we notice that $\beta^+ \geq \beta$ *ie* the intra-community nodes of the graph tend to have more similar labels (*positive curvature edges*) than inter-community (*negative curvature edges*) ones. For heterophilic datasets, using positively curved edges does not improve homophily. According to (Zhu et al., 2020), for heterophilic graphs, the 2-hop is more homophilic than the 1-hop. However, we can observe that according to the curvature-constrained edge homophily using a

two-hop neighborhood is only greater for small datasets *ie WebKB dataset*. Finally, it can be noted that the negative curvature homophily is generally higher than the initial metrics for heterophilic datasets.

### 3.4 Curvature-Constrained Message Passing

Based on the previously introduced curvature-constrained homophily measures, we propose to dissociate the spread of information in the graph based on the Ollivier curvature of the graph edges. We consider diffusions for both one-hop and two-hop connectivity (see the examples in Figure 2).

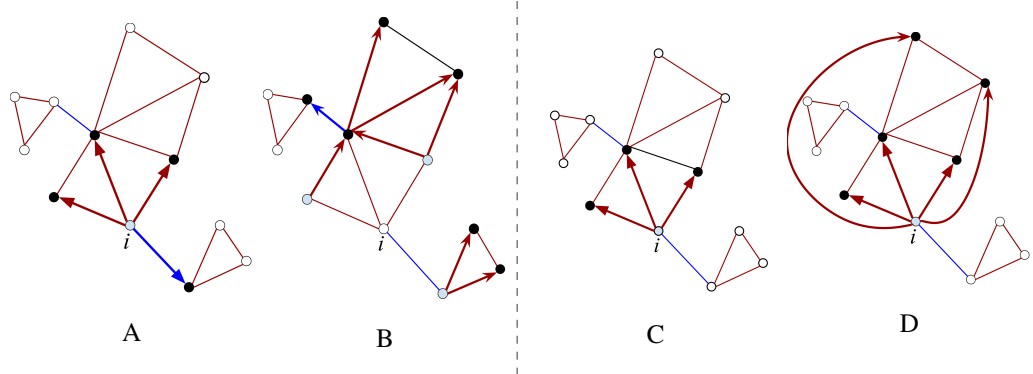

Figure 2: On the left a classic message passing for the first (A) and second layers (B) starting from a given node $i$. On the right an example of one-hop positive curvature (C) and two-hop positive curvature (D) message passing. The message is propagated not only to the adjacent nodes but also to those at distance two along positively curved edges i.e following a chain of positively curved edges of size 2.

We propose to extend the aggregation part of the classic MPNNs 1 :

$$
\begin{aligned}
Curv_{m_i}^{+\,(\ell)} &= \text{AGGREGATE}^{(\ell)}\left(\left\{h_j^{(\ell)} : j \in \mathcal{N}^+(i)\right\}\right) \\
h_i^{\ell} &= \text{UPDATE}^{(\ell)}\left(h_i^{(\ell-1)}, Curv_{m_i}^{+\,(\ell)}\right)
\end{aligned}
\tag{8}
$$

Where $\mathcal{N}^+$ represents the neighborhood of nodes that are connected by a positively curved edge to $i$. Similarly, $Curv_{m_i}^-$ is defined in the same way by considering $\mathcal{N}^-$ instead of $\mathcal{N}^+$.

Propagating information through the curvature of edges offers greater flexibility in learning representation. For a two-layer GNN, we can either only exchange information on edges with negative or positive curvature *i.e.* using one curvature adjacency matrix, or first broadcast information on edges with positive and then negative curvature, or using both curvature adjacency matrix for the two different layers.

**One-hop curvature** We eliminate either the edges with negative curvature or the positive edges, thereby simplifying the graph's connectivity. In this case $W_1(\mu_i, \mu_j)$ of equation 5 decreases; therefore the number of edges which are strongly negatively curved (Topping et al., 2022), responsible for bottlenecks, is reduced. In addition, sparsifying the graph has several advantages, (1) helps to reduce oversmoothing (Rong et al., 2019), (2) we drastically reduce the diameter of the graph, thereby reducing the number of hops needed to join two nodes also helps to prevent over-squashing (Deac et al., 2022). We show empirically that the use of one-hop is beneficial in limiting bottlenecks by noticing an increase in the normalized spectral gap after rewiring.

**Two-hop curvature** Using a neighborhood with multiple hops allows us to mitigate the limitation of classical MPNNs where nodes can only communicate through direct neighbors. Indeed by densifying the graph with multiple hops, we can now transmit information directly with distant nodes

(Brüel-Gabrielsson et al., 2022)(Abboud et al., 2022). This procedure eliminates the requirement to repeat messages across powers of the adjacency matrix thereby reducing the risk of over-squashing (Topping et al., 2022). However, depending on the size of the graph, this can considerably increase the computational cost of the (Gutteridge et al., 2023) method. By working only on a two-hop neighborhood according to a certain curvature of the edges, it is possible to limit the densification of the graph and therefore to reduce the computational cost of the two-hop while facilitating the exchange of information between the distant nodes.

**Using one-hop and two-hop curvature on layers**  Remember that if the distance $k$ between nodes $i$ and $j$ is greater than one, their interaction occurs only at the $k^{th}$ layers. For two-layer GNNs, using a one-hop and then a two-hop curvature between layers allows for faster interaction between distant nodes. This procedure restricts dynamic rewiring message passing (Gutteridge et al., 2023) to only positive or negative curvatures. Using this framework, as demonstrated in (Gutteridge et al., 2023), helps mitigate over-squashing. Restricting this process also addresses one of the paper's limitations, which is that it can only be used for very deep GNN models.

**Positive curvature adjacency matrix**  Diffusing information only on edges with positive curvature allows information to be exchanged only within the communities of the graph. Based on the curvature-constrained homophily used positive curvature adjacency matrix can be useful on homophilic datasets.

**Negative curvature adjacency matrix**  As discussed by Deac et al. (2022) working with negatively curved edges may seem counterintuitive in relation to the recommendation to avoid negatively curved edges (Topping et al., 2022). We confirm the results of Deac et al. (2022) by showing empirically that diffusing information through negatively curved edges improves performance and mitigates the oversquashing phenomenon.

## 4 EXPERIMENTS

### 4.1 DATASETS

We carry out experiments on eleven different datasets for the node classification task of which 7 heterophilic datasets Tang et al. (2009) Rozemberczki et al. (2021)Platonov et al. (2023) and 4 homophilic datasets McAuley et al. (2015) Sen et al. (2008). More details are provided in appendix A.1. The dataset statistics are described in Table 2. We also note the construction time of the curvature matrix according to Ollivier and augmented Forman (AF) (in seconds).

| Dataset | # Nodes | # Edges | # Classes | # Olliver time | # AF-Forman time |
|---------|---------|---------|-----------|----------------|------------------|
| Squirrel | 5021 | 217073 | 5 | $\approx 836$ | $\approx 202$ |
| Chameleon | 2277 | 36101 | 5 | $\approx 11$ | $\approx 9$ |
| Texas | 181 | 309 | 5 | $\approx 1$ | $\approx 1$ |
| Wisconsin | 251 | 499 | 5 | $\approx 1$ | $\approx 1$ |
| Cornell | 181 | 295 | 5 | $\approx 1$ | $\approx 1$ |
| Roman-empire | 22662 | 32927 | 18 | $\approx 75$ | $\approx 2$ |
| Actor | 7600 | 33544 | 5 | $\approx 15$ | $\approx 4$ |
| Citeseer | 3 312 | 4 715 | 6 | $\approx 3$ | $\approx 1$ |
| Cora | 2 708 | 5 429 | 7 | $\approx 2$ | $\approx 1$ |
| Computers | 13752 | 245861 | 10 | $\approx 287$ | $\approx 135$ |
| Photo | 7650 | 119081 | 8 | $\approx 61$ | $\approx 34$ |

Table 2: Description of datasets

### 4.2 BASELINE

We compare our method with four other methods based on rewiring techniques. We provide the results of FA Alon & Yahav (2021) which rewires only the last layer, DIGL (Klicpera et al., 2019)[1],

---

[1]https://github.com/gasteigerjo/gdc

SDRF(Topping et al., 2022)[2] and FOSR(Karhadkar et al., 2023)[3].

We use the hyperparameters that has been defined in the original publication and fine-tune the number of iterations. For DIGL we fine tune top $k$ for $\{8, 16, 32, 64, 128\}$ and $\{0.05, 0.1, 0.15\}$ for the personalized PageRank (Page et al., 1998).

## 4.3 SETUP

For the experiments we use the same framework as (Pei et al., 2020) to evaluate the robustness of each method. Thus, we fix the number of layers to 2, the dropout to $= 0.5$, learning rate to $0.005$, patience of 100 epochs, weight decay of $5E^{-6}$ (Texas/Wisconsin/Cornell) or $5E^{-5}$ (other datasets). The number of hidden states is 32 (Texas/Wisconsin/Cornell), 48 (Squirrel/ Chameleon/Roman-Empire), 32 (Actor) and 16 (Cora/Citeseer) except for Amazon Photo and Computers where the hidden states is 64 and we use a learning rate of 0.01 following the usual framework presented in (Shchur et al., 2018).

We use the two most popular GNNs, GCN (Kipf & Welling, 2017) and GAT (Veličković et al., 2018) as a basis and compare the different methods for rewiring the input graph.

For all the graphs datasets we take a random sample of nodes of $60\%$ for training, and $20\%$ for validation and $20\%$ for testing. We report the average accuracy of each method on 100 random samples.

**Curvature-Constrained Message Passing (CCMP) configuration**  Depending on the curvature method, we use different configurations for the datasets. As the measure of Olliver's curvature is bounded, we consider an adjacency matrix curved negatively/positively respectively in such a way that $e_{ij} \leq 0$ and $e_{ij} \geq 0$. For the augmented Forman curvature measure we consider an adjacency matrix curved negatively/positively respectively in such a way that the curvature is less/more important than the average of the curvature of the edges on the graph. Details of the adjacency matrix types for the different datasets are presented in appendix A.3. We designate the use of the Olliver curvature with $\text{CCMP}_O$ and the augmented Forman curvature with $\text{CCMP}_A$.

## 4.4 RESULTS

| Backbone | Method | Cora | Citeseer | Amazon Photo | Amazon Computers |
|---|---|---|---|---|---|
| GCN | None | 87.73 ±0.25 | 76.01 ±0.25 | 89.89 ±0.37 | 80.45 ±0.56 |
| | DIGL | **88.22** ±0.28 | 76.18 ±0.34 | **90.31** ±0.43 | **83.04**±0.43 |
| | FA | 29.86 ±0.28 | 22.31 ±0.34 | OOM | OOM |
| | SDRF | 87.73 ±0.31 | 76.43 ±0.32 | $\geq 48H$ | $\geq 48H$ |
| | FOSR | 87.94±0.26 | 76.34 ±0.27 | 90.24 ±0.31 | 80.78 ±0.43 |
| | **CCMP**$_0$ | 87.34 ±0.29 | **76.68** ±0.28 | 89.94 ±0.29 | 81.66 ±0.47 |
| | **CCMP**$_A$ | 85.60 ±0.37 | 75.76 ±0.39 | 90.31 ±0.38 | 81.84±0.45 |
| GAT | None | 87.65 ±0.24 | 76.20 ±0.27 | 88.76 ±0.39 | 80.72(2.5) ±0.53 |
| | DIGL | **88.31** ±0.29 | 76.22 ±0.34 | **90.32**±0.46 | **83.28** ±0.49 |
| | FA | 30.44 ±0.26 | 23.11 ±0.32 | OOM | OOM |
| | SDRF | 88.11 ±0.28 | 76.26 ±0.31 | $\geq 48H$ | $\geq 48H$ |
| | FOSR | 88.13 ±0.27 | 75.94±0.32 | 90.12 ±0.41 | 80.78 |
| | **CCMP**$_O$ | 84.59±0.30 | **76.44**±0.33 | 89.47 ±0.37 | 81.41 ±0.47 |
| | **CCMP**$_A$ | 86.16 ±0.32 | 75.88 ±0.44 | 89.88 ±0.22 | 81.96 ±0.51 |

Table 3: Experimental results on **homophilic** datasets. Best score in bold and second best score underlined.

Tables 3, 4 and 5 show the results of our experiments. The rewiring approaches to limit over-squashing SDRF, FORSR, and CCMP produce quite comparable results for homophilic datasets. Indeed in such datasets nearby neighborhood information is sufficient to achieve good performance. DIGL tends to improve connectivity between nodes with short diffusion. As a result, DIGL's superior performance can be explained by the fact that it will add positively curved edges. This

---

[2]https://github.com/jctops/understanding-oversquashing/tree/main
[3]https://github.com/kedar2/FoSR/tree/main

|  | Base (GCN) | DIGL | FA | SRDF | FOSR | $CCMP_o$ | $CCMP_a$ |
|---|---|---|---|---|---|---|---|
| **Cham.** | 65.35±0.54 | 54.82 ±0.48 | 26.34 ±0.61 | 63.08 ±0.37 | **67.98** ±0.40 | 63.22 ±0.45 | 65.66 ±0.44 |
| **Squir.** | 51.30±0.38 | 40.53 ±0.29 | 22.88 ±0.42 | 49.11±0.28 | 52.63 ±0.30 | 53.36 ±0.22 | **54.79** ±0.31 |
| **Actor** | 30.02±0.22 | 26.75 ±0.23 | 26.03±0.30 | 31.85 ±0.22 | 29.26±0.23 | 33.57±0.22 | **34.59**±0.24 |
| **Texas** | 56.19 ±1.61 | 45.95 ±1.58 | 55.93 ±1.76 | 59.79 ±1.71 | 61.35 ±1.25 | 64.67 ±1.81 | **69.67** ±1.64 |
| **Wisc.** | 55.12±1.51 | 46.90 ±1.28 | 46.77±1.48 | 58.49 ±1.23 | 55.60 ±1.25 | 66.40±1.24 | **67.80** ±1.49 |
| **Corn.** | 44.78 ±1.45 | 44.46 ±1.37 | 45.33±1.55 | 47.73 ±1.51 | 45.11 ±1.47 | **58.91** ±1.82 | 58.54 ±1.57 |
| **R-emp.** | 51.66 ±0.17 | 53.93 ±0.14 | OOM | 52.53 ±0.13 | 52.38 ±0.21 | 58.58 ±0.14 | **58.91** ±0.19 |

Table 4: Experimental results on **heterophilic** datasets with **GCN** as backbone. Best score in bold and second-best score underlined.

|  | Base (GAT) | DIGL | FA | SRDF | FOSR | $CCMP_o$ | $CCMP_a$ |
|---|---|---|---|---|---|---|---|
| **Cham.** | 65.07 ±0.41 | 56.34 ±0.43 | 27.11 ±0.56 | 63.15±0.44 | **66.61** ±0.45 | 63.09 ±0.52 | 65.59 ±0.43 |
| **Squi.** | 50.87 ±0.56 | 41.65 ±0.68 | 21.49 ±0.71 | 50.36 ± 0.38 | 52.02 ±0.43 | 51.82 ±0.32 | **54.74** ±0.52 |
| **Actor** | 29.92 ±0.23 | 31.22 ±0.47 | 28.20 ±0.51 | 31.47 ±0.25 | 29.73 ±0.24 | 33.23 ±0.22 | **34.23** ±0.23 |
| **Texas** | 56.84 ±1.61 | 46.49 ±1.63 | 56.17 ±1.71 | 57.45 ±1.62 | 61.85 ±1.41 | **71.78** ±1.39 | 70.65 ±1.36 |
| **Wisc.** | 53.58 ±1.39 | 46.29 ±1.47 | 46.95 ±1.52 | 56.80 ±1.29 | 54.06±1.27 | 65.77±1.32 | **68.59** ±1.41) |
| **Cornell** | 46.05 ±1.49 | 44.05 ±1.44 | 44.60 ±1.74 | 48.03 ±1.66 | 48.30±1.61 | **60.43** ±1.47 | 59.81 ±1.49 |
| **R-Emp.** | 49.23 ±0.33 | 53.89 ±0.16 | OOM | 50.75 ±0.17 | 49.54 ±0.31 | **57.36** ±0.19 | 56.78 ±0.39 |

Table 5: Experimental results on **heterophilic** datasets with **GAT** as backbone. Best score in bold and second-best score underlined.

rewiring process allows for the improvement of homophily on homophilic datasets illustrated by our curvature-constrained measure $\beta^+$.

Tables 4 and 5 show that for 6 among 7 heterophilic datasets our method achieves the best results. Of all these datasets on average, $CCMP_O$ and $CCMP_A$ outperforms the original adjacency matrix with a base of GCN and GAT by 14.24% and 16.55%. Note that SRDF, FOSR, and CCMP outperform DIGL on heterophilic datasets because neighboring nodes tend to have different labels.

Here we present some results based on $CCMP_O$. According to (Topping et al., 2022), we are interested in the original graphs' strongly negatively curved edges. After removing these edges in Wisconsin and Cornell's datasets, the first deciles of the curvature change from $-0.33, -0.33, -0.25$ in the original adjacency matrix to $+0.37, +0.33, +0.06$ in the two hop negatively curved adjacency matrix.
Using a one-hop curvature allows to reduce the size of the graph. Consequently, on Squirrel, Actor, and Roman-Empire, the computational cost is reduced from 10% to 40%. Besides, the fact that the normalized spectral gap increases from 5% to 87% on these datasets shows that using a one-hop curvature allows to mitigate over-squashing.

## 5 CONCLUSION

In this paper, we present a method, that can be applied to any MPNN architecture, to distribute messages according to the curvature of the graph's edges. This allows to mitigate over-squashing, one of the main drawbacks of classical MPNN architectures. Taking into account a new curvature-constrained homophily measure, we developed various variants of the method to propagate the information along curved edges: propagation along negative or positive edges, with one or two hops. The experiments show a significant improvement over comparable state-of-the-art methods based on rewiring, empirically proving the utility of curvature-based constraining of message passing for the reduction of bottlenecks. In future works, we plan to consider other curvature measures. We will also study the effect of using very deep GNN models with different curvature adjacency matrices for long-range graph benchmarks.

# A APPENDIX

## A.1 DATASETS

In this section, we present the dataset information for the experiments.

**WebKB**: in this dataset, nodes represent web pages, and edges are hyperlinks between them. Node features are the bag-of-words representation of web pages.

**Actor**: Each node corresponds to an actor, and one edge between two nodes denotes co-occurrence on the same Wikipedia page (Tang et al., 2009). Node features correspond to some keywords in the Wikipedia pages.

**Wikipedia network**. Nodes represent web pages and edges are mutual links between pages (Rozemberczki et al., 2021). Node features correspond to informative nouns in the Wikipedia pages.

**Roman-empire**. Graph-of-Words: each node corresponds to one word from the Roman Empire article from English Wikipedia Platonov et al. (2023). Two words are connected with an edge if either these words follow each other in the text, or these words are connected in the dependency tree of the sentence.

**Scientific publication networks**. Cora and Citeseer datasets Sen et al. (2008) describe citations to scientific publications. Each publication is described by a one-hot vector indicating whether a word is absent/present in the publication abstract.

**Amazon**. An extract of the Amazon purchase graph McAuley et al. (2015), where nodes represent goods, and edges indicate whether two goods are frequently purchased together. The features are the bags of words of the product descriptions.

## A.2 DETAILS ABOUT CURVATURE-CONSTRAINED HOMOPHILY

Here we specify the details of the Curvature-Constrained homophily used in the experiments for our rewiring methods for layers 1/2.

|  | **Dataset** | $\beta$ | $\beta$-CCMP$_O$ | $\beta$-CCMP$_A$ |
|---|---|---|---|---|
| | Squirrel | 0.23/0.23 | 0.28/0.28 | 0.26/0.26 |
| | Chameleon | 0.26/0.26 | 0.29/0.32 | 0.32/0.32 |
| | Texas | 0.31/0.31 | 0.47/0.47 | 0.56/0.56 |
| Heterophilic | Wisconsin | 0.36/0.36 | 0.38/0.38 | 0.42/0.42 |
| | Cornell | 0.34/0.34 | 0.40/0.40 | 0.39/0.39 |
| | R-empire/ | 0.29/0.29 | 0.48/0.48 | 0.33/0.33 |
| | Actor | 0.32/0.32 | 0.73/0.73 | 0.64/0.64 |
| | Cora | 0.84/0.84 | 0.83/0.95 | 0.84/0.98 |
| Homophilic | Citeseer | 0.81/0.81 | 0.84/0.86 | 0.89/0.89 |
| | Photo | 0.84/0.84 | 0.94/0.94 | 0.89/0.89 |
| | Computers | 0.78/0.78 | 0.93/0.93 | 0.83/0.83 |

Table 6: Comparison of edge homophily measures. The last column reports the max gain in homophily obtained by using the curvature-constrained edge homophily as opposed to edge homophily.

## A.3 DETAILS OF CONFIGURATION FOR CCMP$_O$ AND CCMP$_A$

The optimal configurations for CCMP$_O$ is: (1) for Cora and Citeseer, for the first layer a one-hop negatively curved adjacency matrix and for the second layer a one-hop positively curved adjacency matrix, (2) for small datasets like Texas, Wisconsin and Cornell, a two-hop negatively curved adjacency matrix for both layers. (3) for Squirrel and Roman-Empire datasets, a negatively curved one-hop adjacency matrix on two layers, (4) for Actor and Amazon computers, a positive one-hop adjacency matrix on two layers, (5) for Chameleon and Amazon photo, for the first layer a one-hop positive adjacency matrix and for the second layer a two-hop positive adjacency matrix.

For CCMP$_A$: (1) Chameleon, Squirrel, Citeseer, Amazon Computers, Photo we use a positive one-hop adjacency matrix on two layers, (2) For Actor and Romain-empire we use a negatively curved

one-hop adjacency matrix on two layers, (3) For the other datasets the configuration is the same as for the CCMP$_O$.

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
