# OpenReview forum: "Curvature MPNNs : Improving Message Passing with Local Structural Properties"
_ICLR.cc/2024/Conference — Submitted to ICLR 2024_

### Official Review · Reviewer_6GYo · 2023-10-29

**Soundness:** 2 fair
**Presentation:** 2 fair
**Contribution:** 2 fair
**Rating:** 5
**Confidence:** 3

**Summary:**

This paper suggests a new GNN architecture that uses the curvature of the input graph to determine which nodes can pass messages. Specifically, the paper suggests passing messages only along the negative curved edges. The paper then shows that this improves performance for hetrophilic datasets.

**Strengths:**

The idea of passing messages based on the curvature of the edges is interesting.

The idea is simple and seems to work well and have been tested against quite a few appropriate methods on a variety of different graphs.

**Weaknesses:**

The writing of the paper, specially the first 7 pages contain many mistakes. Please see the questions section for many of these. Specifically the math presented in the paper has many error (definition of $W_1$, sparsifying the graph improving the diameter, and other statements such as that).

There also many formatting issues, I list some of them in the questions section. I would highly recommend the authors to fix these.

However, I think many of these are fixable and I am very willing to increase my score on seeing a fixed revision.

**Questions:**

Formatting issue between 2nd and 3rd paragraphs.

Instead of using a fake link, I recommend you GitHub anonymous. It creates an anonymous version of GitHub repositories.

On page 3 you say the formation of curvature in Topping et al. is to improve expressiveness. Could you elaborate? My understanding was that it was to improve computability.

The formatting for Banerjee, Kardhakar, and Montufar 2023 makes it look like two separate papers above equation 2, and drops the name of the first author Banerjee in other places. This needs to be fixed.

Just above equation 4, $\pi$ is not a measure for this paper. Also I would recommend not using $\pi$ to represent a scalar different from the constant $\pi$. The prior work cited uses $\alpha$.

For equation 5, you are missing the restrictions on the marginals of $M$ as well the restriction that $M$ is a probability measure on the product space.

How is the two hop $\beta^+$ defined?

I also do not understand how Figure 2D is the 2hop *positive message passing*. What do the dashed lines represent? The message also propagated further for the regular message passing framework. Which is contrary to the caption of Figure 2.

For 1 hop message passing. It says that sparsifying the graph **reduces** the diameter of the graph. This is just blatantly false. For example, for the graph graph used, removing either the positive or negatively curved edges results in the graph being disconnected which makes the diameter infinite.

---

> ### Author Response · Authors · 2023-11-17
>
> Thank you for reviewing our paper and for the insightful comments. We hope our answers to the questions will address the concerns and clarify the contributions of the paper.
>
> > There also many formatting issues, I list some of them in the questions section. I would highly recommend the authors to fix these.
>
> * We agree that the diameter of the graph can become infinite. We have modified the sentence accordingly.
> * We have fixed all the others mentioned mistakes and formatting issues.
>
> > On page 3 you say the formation of curvature in Topping et al. is to improve expressiveness. Could you elaborate? My understanding  was that it was to improve computability.
>
> The curvature proposed in [1] is more expressive than Forman Curvature since it  takes into account triangles and cycles of size 4 Section 3 equation 3. Furthermore it’s can be use to improve computability on very large graphs w.r.t Ollivier Curvature.
>
> >How is the two hop defined?
>
> * The two-hop neighbourhood is defined using a neighbourhood of size 2 only on edges with certain curvatures. In the case of two-hop positive curvature, the information is broadcast both to neighbours connected by edges with positive curvature and to neighbours of neighbours connected in the same way by edges with positive curvature i.e follows a chain of positively curved edges of size 2.
> We specify this answer in the section 3.4 for greater clarity.
>
>
> > I also do not understand how Figure 2D is the 2hop positive message passing. What do the dashed lines represent? The message also propagated further for the regular message passing framework. Which is contrary to the caption of Figure 2.
>
> * We are modifying Figure 2 and the caption to make it easier to read.
>
> ---
> * We added experiments using other curvature Augmented Forman curvature ((Samal et al 2018) (named CCMP_A in the result table compared to the one initially shown as CCMP_Ollivier). As  CCMP_Ollivier, CCMP_A  significatively improves state of the art methods. We recall the definition of Augmented Forman curvature in Section 3.2 and we provide the details of the configuration for CCMP_A in Appendix A.3. We also indicate the computational cost of the method in table 2.
>
> * We also fixed the broken English sentences.
>
> We would like to once again  thank the reviewers for their great questions and hope that our answers will help them improve their scores

---

> ### Comment · Reviewer_6GYo · 2023-11-18
> **Thank you**
>
> I thank the authors for revising their paper. Based on their revisions, I will increase my score to a 5. However, there are still formatting issues in section 4; one of the tables doesn't have a bottom line. The presentation of the method could also be improved.
>
> In general, for the authors, I think the idea of using curvature to measure homophily is interesting, and for heterophilic datasets, the method does seem to improve things. These are two positives such that if the paper were well written I think could be accepted at a good venue.

---

> > ### Author Response · Authors · 2023-11-22
> >
> > Thank you for your comments and suggestions. We will take all this information into account as we progress with this work.

---

### Official Review · Reviewer_YMEu · 2023-10-30

**Soundness:** 3 good
**Presentation:** 2 fair
**Contribution:** 2 fair
**Rating:** 5
**Confidence:** 4

**Summary:**

The authors tackle a significant challenge within Graph Neural Networks (GNNs). In recent years, numerous critiques have arisen regarding the effectiveness of GNNs, and various research endeavors have been dedicated to pinpointing the primary flaws in their message-passing framework. Recent investigations have pointed to concerns like over-smoothing and over-squashing issues. Additionally, recent research has proposed that Ricci curvature can be utilized to pinpoint bottleneck areas within graphs, which are responsible for the over-squashing problem. In this study, the authors set out to leverage Ricci curvature as a means to address these problems. Through a series of experiments, the authors substantiate the effectiveness of their methodology in improving node classification, especially in heterophilic graphs.

**Strengths:**

1. This paper focuses on leveraging the inherent local structural features of graphs within a message-passing framework for Graph Neural Networks (GNNs) by incorporating curvature. The primary objective is to tackle issues such as over-smoothing and over-squashing that are prevalent in GNNs. The idea is novel, and seems to improve the performance, especially in heterophillic graphs.

2. The comparative analysis in Section 3.2, which examines homophily versus curvature, provides valuable insights for the model.

3. By conducting a series of experiments, the authors establish the effectiveness of their methodology in enhancing node classification, with a particular emphasis on its performance in heterophilic graphs.

**Weaknesses:**

1. While most of the paper is written well, Section 3.3 needs to be clarified. If I understand it right, you have constructed five distinct models and conducted experiments with each model on separate datasets. To enhance clarity, consider labeling each model  like Model 1, Model 2, and so forth. Additionally, in the experiments section, explicitly specify which model corresponds to each set of experiments. Without this clarification, there is a potential for misinterpretation, where one might incorrectly assume that there is only one model and that the results represent the performance of a single model.

2. Because of several different models, and separate experiments, it is not clear what is working and what is not.

3. A performance comparison with SOTA GNNs (not just rewiring baselines) would be better to evaluate the performance of the model.

**Questions:**

1. Although there are results suggesting similarities between the expanded Forman Ricci curvature and the Ollivier Ricci curvature, do you anticipate achieving comparable outcomes if you were to substitute your method with the Forman Ricci curvature? Given its computational efficiency, this alternative might be more suitable for handling large graphs.

2. Did you try to use curvature values (positive and negative) directly as weight in message-passing framework, instead of using them to eliminate some edges?

---

> ### Author Response · Authors · 2023-11-17
>
> Thank you for reviewing our paper and for the insightful comments. We hope our answers to the questions will address the concerns and clarify the contributions of the paper.
>
> > Although there are results suggesting similarities between the expanded Forman Ricci curvature and the Ollivier Ricci curvature, do you anticipate achieving comparable outcomes if you were to substitute your method with the Forman Ricci curvature? Given its computational efficiency, this alternative might be more suitable for handling large graphs.
>
> * We added experiments using other curvature Augmented Forman curvature ((Samal et al 2018) (named CCMP_A in the result table compared to the one initially shown as CCMP_Ollivier). As  CCMP_Ollivier, CCMP_A  significatively improves state of the art methods. We recall the definition of Augmented Forman curvature in Section 3.2 and we provide the details of the configuration for CCMP_A in Appendix A.3. We also indicate the computational cost of the method in table 2
>
>
> > Did you try to use curvature values (positive and negative) directly as weight in message-passing framework, instead of using them to eliminate some edges?
>
> * No. However, we plan to use curvature inside an attention mechanism to take into account both the features and the local structural property of the graph.
> ---
>
> We also modified Figure 2 to make it easier to read and fixed the broken English sentences.
>
> We would like to once again  thank the reviewers for their great questions and hope that our answers will help them improve their scores

---

> > ### Comment · Reviewer_YMEu · 2023-11-18
> >
> > Thank you for the new experiments involving Forman Ricci and revisions. It's clear that Forman curvature notably diminishes the time involved, and its performance aligns closely with that of Ollivier Ricci. Nonetheless, my reservations regarding clarity and other shortcomings persist. I concur with fellow reviewers on the need for comprehensive restructuring and proofreading. Enhancing the experimental section's clarity to distinctly outline the utilization of curvature MPNN models from the 5 approaches, and providing a more nuanced discussion comparing these approaches to delineate their effectiveness, would greatly elevate the paper's quality.

---

> > > ### Author Response · Authors · 2023-11-22
> > >
> > > Thank you for your comments and suggestions. We will take all this information into account as we progress with this work.

---

### Official Review · Reviewer_Z4u7 · 2023-10-31

**Soundness:** 1 poor
**Presentation:** 2 fair
**Contribution:** 2 fair
**Rating:** 3
**Confidence:** 4

**Summary:**

This paper tackles two major problems in standard message-passing neural networks on graphs: oversquashing and oversmoothing. For this, they propose to send messages only along edges that have positive curvature and also to increase the neighborhood of nodes by also considering multi-hop neighbours. This new message passing is tested on node classification tasks on homophilic and heterophilic where a gain for heterophilic graphs is reported

**Strengths:**

- Try to tackle the problem of oversmoothing and oversquashing jointly
- Paper is easily to follow

**Weaknesses:**

- The authors often do not back up their results. For example, they claim that  "sparsifying the graph has several advantages, (1) helps to reduce oversmoothing..." While there is a reference to DROPEDGE, it is not clear that sparsifying by removing negatively curved edges reduces oversmoothing. Oversmoothing has a mathematical definition, thus it would be beneficial to prove or show experiments that back these claims up.
- Some passages and "results" seem unrelated or at least not well discussed. For example, in Table 1 the authors show the homophily gains by considering a different computational graph (that has only negatively or positively curved edges). How does this relate to the remainder of the work?
- There are many details missing, e.g., the model itself in Equation 7 doesn't include the multi-hop propagation.
- The experimental section is weak. Only one hyperparameter configuration is used. There should be many hyperparameters tested. Otherwise there is also no need for having a validation set, and also it may be that the baseline methods are not well evaluated. For instance, other papers ([1,2]) report much higher numbers.
- Also regarding the experiments: As the work claims to tackle oversmoothing, it may be beneficial to compare the method to other works that tackle oversmoothing, see, e.g., [1-5]
- While the authors refer to (Pei et al., 2020) for their experimental setup, they do not use the setup therein. For instance, there exist standard splits which makes comparison to other methods easier.
- The overall writing and grammatic should be checked again

**Questions:**

- The authors identify (in accordance with related work) that negatively curved edges may be bottlenecks leading to oversquashing; but then messages are only passed along negatively curved edges. How does this go hand in hand?
- Regarding the experiments: It is not clear how the experiments relate to the rest of the work. For instance, do any of these datasets actually suffer from oversquashing?

While the tackled problem is relevant and interesting, I believe the paper is not ready for publication. It would benefit a lot from backing up more of their claims; either through experiments or mathematical proofs.

[1] Choi, Jeongwhan, et al. "GREAD: Graph Neural Reaction-Diffusion Equations." arXiv preprint arXiv:2211.14208 (2022).
[2] Maskey, Sohir, et al. "A Fractional Graph Laplacian Approach to Oversmoothing." arXiv preprint arXiv:2305.13084 (2023).
[3] Rusch, T. Konstantin, et al. "Graph-coupled oscillator networks." International Conference on Machine Learning. PMLR, 2022.
[4] Pei, H., Wei, B., Chang, K. C. -C., Lei, Y., and Yang, B. (2019). “Geom-GCN: Geometric Graph Convolutional Networks”. In: International Conference on Learning Representation
[5] Yan, Y., Hashemi, M., Swersky, K., Yang, Y., and Koutra, D. (2022). “Two Sides of the Same Coin: Heterophily and Oversmoothing in Graph Convolutional Neural Networks”. In: 2022 IEEE International Conference on Data Mining

---

> ### Author Response · Authors · 2023-11-17
>
> Thank you for reviewing our paper and for the insightful comments. We hope our answers to the questions will address the concerns and clarify the contributions of the paper.
>
> >  The authors identify (in accordance with related work) that negatively curved edges may be bottlenecks leading to oversquashing, but then messages are only passed along negatively curved edges. How does this go hand in hand?
>
> * Indeed [1] shows that it is the negatively curved edges are responsible for the over-squashing phenomenon.
> By removing all the positively curved edges, the curvature of the nodes and their neighbors is updated. So the obtained graph does not have only negatively curved edges.
> We can see a considerable increase in the normalised spectral gap when messages are transmitted only along edges with negative curvature.
> For exemple on Romain Empire using a negatively curved one-hop adjacency matrix improve the spectral gap by 87%.
> In addition, we find that on heterophilic datasets the measure of curvature-constrained homophily increases, i.e. nodes that are connected by an edge with negative curvature tend to have the same label.
>
> > Regarding the experiments: It is not clear how the experiments relate to the rest of the work. For instance, do any of these datasets actually suffer from oversquashing?
>
>
> * Long-range dependencies are more relevant in heterophilic datasets because local information is not sufficient.That's why reducing bottlenecks is so important for facilitating long-distance information exchange.
> This observation is largely confirmed by [1] and therefore, methods that aim to reduce bottlenecks  are tested on heterophilic datasets.
>
>
> [1]  UNDERSTANDING OVER-SQUASHING AND BOTTLENECKS ON GRAPHS VIA CURVATURE Topping et al 2022
>
> ---
> * We added experiments using other curvature Augmented Forman curvature ((Samal et al 2018) (named CCMP_A in the result table compared to the one initially shown as CCMP_Ollivier). As  CCMP_Ollivier, CCMP_A  significatively improves state of the art methods. We recall the definition of Augmented Forman curvature in Section 3.2 and we provide the details of the configuration for CCMP_A in Appendix A.3. We also indicate the computational cost of the method in table 2.
>
> * We also fixed the broken English sentences.
>
> We would like to once again  thank the reviewers for their great questions and hope that our answers will help them improve their scores

---

> > ### Comment · Reviewer_Z4u7 · 2023-11-22
> >
> > Thank you for the response. However, most of my questions remain unanswered. For example, the authors still test only on graphs where long-range interactions may be important; thus comparing against methods that can model long-range interactions (not only rewiring methods but also methods that solve oversmoothing, etc.) would be important. Also, the clear connection between theoretical motivation and experimental results is still hand-wavy. Therefore, I keep my score.

---

### Official Review · Reviewer_8f8F · 2023-11-01

**Soundness:** 2 fair
**Presentation:** 1 poor
**Contribution:** 1 poor
**Rating:** 1
**Confidence:** 2

**Summary:**

The authors use Olivier curvature to modify message passing in MPNNs. More specifically, message passing can now be controlled using neighbours connected by positively or negatively curved edges only. Since curvature is somewhat related to homophily and heterophily, this extended message passing scheme provides better adjustment to  homophily vs heterophilic datasets.

**Strengths:**

1. The integration of curvature into the message passing is a nice idea.
2. The approach shows promise empirically.

**Weaknesses:**

1. The paper is very badly written. Many sentences do not parse well.
2. The idea is nice, but it is extremely simple. The overall technical depth is not on par with one expects from an ICLR paper.
3. The presentation of the different ways that curvature can be used (section 4.4.) could have been more precise.
4. There is no theoretical justification of the proposed method.
In summary,  nice idea but not enough substantial contributions.

**Questions:**

Q1. Could you explain how one should choose between message passing between positive vs. negative curved neighbourhoods? Or does one  simply tries all possible combinations and picks the best?

---

> ### Author Response · Authors · 2023-11-17
>
> Thank you for reviewing our paper and for the comments. We hope our answers to the questions will address the concerns and clarify the contributions of the paper.
>
> - When the dataset is homophilic, the best strategy is to propagate information inside the communities i.e. using positively curved edges (Table 1 and Table 3 shows that this strategy has been empirically confirmed on the 4 homophilic datasets). For heterophilic datasets, the experimentations has shown that the best results can generally be obtained when the strategy is choosing according to the higher value of the curvature constrained edge homophily  (c.f. see appendix A.2 and A3).
>
> -We added experiments using the Augmented Forman curvature (Samal et al 2018)  (named CCMP_A in the result table compared to the one initially shown as CCMP_Ollivier). As  CCMP_Ollivier,CCMP_A  significatively improves baseline. Furthermore, by comparing running times we could also observe a significant computational gain.
>
> -We also fixed the broken English sentences.

---

> > ### Comment · Reviewer_8f8F · 2023-11-18
> > **Response to authors**
> >
> > Thank you for the response. I appreciate the additional guidance given as to when to chose positive vs negative curvature. Despite the changes made, I am still not very pleased with the current writeup. A more detailed presentation of the different methods (3.4) is needed, for example. See also comments by other reviewers.

---

> > > ### Author Response · Authors · 2023-11-22
> > >
> > > Thank you for your comments and suggestions. We will take all this information into account as we progress with this work.

---

### Meta-Review · Area_Chair_2P1z · 2023-12-06

**Metareview:**

This paper applies graph discrete curvature (Forman, Ollivier) to build a message-passing network, which propagates based on the sign of the edge curvature. The authors showed preliminary results based on Ollivier's discrete Ricci curvature that can improve performance on heterophilic graphs.

Strengths:

- A nice idea of applying discrete curvature into GNNs that aims to tackle the problem of over-smoothing and over-squashing jointly.

Weaknesses:

- The writing quality including English, Math, and overall clarity, is not good enough.

- The author's claims are not well supported by formal analysis or a thorough experimental study. Theoretically, there is no in-depth analysis, or enough intuitions, of the discrete curvature or why it helps the over-smoothing/over-squashing problems. This is in contrast to the advanced mathematical tools the authors have used. The introduction of discrete curvature can be improved. Experimentally, the current evaluation is not complete. Based on the reviewers, the authors could compare against SOTA baseline methods that tackle oversmooth, and clarify the hyperparameter configurations.

We thank the authors for their efforts and acknowledge the potential value of this work. We encourage the authors to revise based on the reviewers' comments for its future development.

**Justification For Why Not Higher Score:**

All reviewers suggest a major restructuring/rewriting and further improvement of the experimental evaluation.

**Justification For Why Not Lower Score:**

N/A

---

### Decision · Program_Chairs · 2024-01-16

Reject